# Rootstocks with Different Tolerance Grade to Citrus Tristeza Virus Induce Dissimilar Volatile Profile in *Citrus sinensis* and Avoidance Response in the Vector *Aphis gossypii* Glover

**DOI:** 10.3390/plants11243426

**Published:** 2022-12-08

**Authors:** Salvatore Guarino, Francesco Mercati, Sergio Fatta Del Bosco, Antonio Motisi, Loredana Abbate

**Affiliations:** Institute of Biosciences and Bioresources (IBBR), National Research Council of Italy (CNR), Corso Calatafimi 414, 90129 Palermo, Italy

**Keywords:** CTV, VOCs, cotton aphid, virus vector, *Citrus aurantium*, Carrizo citrange, Forner-Alcaide no. 5, Volkamer lemon

## Abstract

The citrus tristeza virus (CTV) is an agent of devastating epidemics of the citrus plant grafted on *Citrus aurantium,* one of the main rootstocks still used in the Mediterranean area. Consequently, CTV-tolerant alternative citrus rootstocks are considered necessary to manage this disease and/or its vector; that in Mediterranean countries is the aphid *Aphis gossypii*. In this study, we analyzed the VOCs emitted from *Citrus sinensis* plants grafted on the CTV-susceptible *C. aurantium* and on the CTV-tolerant Volkamer lemon, Forner-Alcaide no. 5, and Carrizo citrange. Furthermore, the aphid preference/avoidance response toward these combinations was evaluated in a semi-field experiment. The VOC profiles recorded on the leaves of *C. sinensis* grafted on the four rootstocks listed above showed significant differences in the abundances and ratios of the compounds emitted. The behavioral experiments indicated that *A. gossypii* prefers to orient and establish on the *C. sinensis* plants grafted on *C. aurantium* rather than on that grafted on the three CTV-tolerant varieties. The possibility that this avoidance mechanism is triggered by the different profile of the VOC emitted by the different combinations and the consequent susceptibility/tolerance shown toward CTV is discussed.

## 1. Introduction

The citrus tristeza virus (CTV) is an aphid-borne closterovirus (genus Closterovirus, family Closteroviridae), an agent of devastating epidemics that drastically modified the course of the citrus industry [1]. CTV infection causes a decline syndrome of different citrus species such as sweet oranges (*Citrus sinensis* L.), mandarins (*C. reticulata* Blanco), and grapefruits (*C. paradisi* Macf.), propagated on sour orange (*C. aurantium* L. Osb.) [1,2]. The sour orange was formerly the most important rootstock in almost every citrus-growing region worldwide [3] and still is highly widespread in the countries of the Mediterranean basin. The CTV epidemics determined the death of almost 100 million trees propagated on sour orange and became a limit for their use as a rootstock [1,4]. CTV attacks on the trees grafted on sour orange induce a quick plant decline, which results from a virus-induced graft incompatibility between the scion and rootstock [5].

This generated the necessity of finding alternative CTV-tolerant rootstocks that, similarly to sour orange, produce good yields of high-quality, tolerate abiotic stress factors, such as frost and salinity, and resist biotic constraints, such as gummosis and foot rot caused by *Phytophthora* spp. and the viroids of citrus exocortis (CEVd) and hop stunt (HSVd) [6,7].

In consideration of the efficient CTV dispersals by vectors, the propagation of citrus on CTV-tolerant rootstocks is nowadays considered the only viable option to manage CTV and avoid the tristeza decline [8,9].

In this context, among the CTV-tolerant rootstock candidates to substitute sour orange, only a few genotypes have shown promising potential. Among them, there are Volkamer lemon (*C. volkameriana* Ten. & Pasq.), the hybrids Carrizo citrange (*C*. *sinensis* × *Poncirus trifoliata* (L.) Raf.), and Forner-Alcaide no. 5 (*C. reshni* Hort. Ex Tan. × *P. trifoliata*) [10,11,12,13,14,15].

A recent study carried out from Guarino et al. [16] evidenced that the leaf volatile organic compound (VOC) profiles recorded in the three CTV-tolerant rootstocks, listed above, show marked qualitative and quantitative differences in comparison with those of the CTV-susceptible *C. aurantium*. In the same study, it has been speculated that these differences in the emitted VOCs can be involved in the tolerance mechanism evidenced in Volkamer lemon, Carrizo citrange, and Forner-Alcaide no. 5 by deterring the CTV vectors, in consideration of the importance that such secondary metabolites have in the trophic systems plants—herbivores [16]. However, the same study did not clarify if the rootstocks can induce different VOCs in the grafted scions or if the CTV vectors elicit a different preference behavior toward the species testes.

It is well known that CTV is transmitted by several aphid species in a semipersistent mode with variable effectiveness [17,18]. The most efficient vector species, *Toxoptera citricida* (Kirkaldy) [19], is widespread in Asia, Australia, sub-Saharan Africa, Central and South America, and different Caribbean countries [20,21,22]. However, this aphid is not present in Mediterranean regions apart from rare detections in isolated citrus trees in northern Spain and Portugal, far from the important citrus-producing areas [23]. In the Mediterranean basin and areas of North America, the main CTV vector is considered *Aphis gossypii* (Glover), also known as the cotton-melon aphid [24,25,26,27]. *Aphis gossypii* is a highly polyphagous aphid species [28] and it is an extremely problematic pest because it can develop a high population density in a short period of time and exhibit insecticide resistance [28,29,30].

Similarly to the majority of herbivore insects, this aphid relies on VOCs to orient toward the preferred host plant species [31]. Consequently, the different VOC emitted from the host plant can contribute to determine the mechanism of host plant finding/acceptance or avoidance/escaping. In consideration of the importance of *A. gossypii* as the main CTV vector in Mediterranean countries and North America, the role of the evaluation of VOCs in citrus, that can be attacked by this pest, is of crucial importance.

Therefore, the purpose of this study is to clarify if the distinctive scion/rootstock combinations can induce a preference or avoidance behavior in *A. gossypii*, and if CTV-tolerant rootstocks can modulate the VOC emission in *C. sinensis* scions, inducing different profiles in comparison to the susceptible sour orange. Overall, the specific objectives of the present work were: (i) to assess if the CTV vector *A. gossypii* exhibits specific behavior toward plants grafted on CTV-susceptible and CTV-tolerant rootstocks and (ii) to evaluate the influence of the rootstocks in *C. sinensis* plants in terms of VOCs emitted from leaves.

## 2. Results

### 2.1. Behavioral Bioassays

The response of *A. gossypii* individuals toward the *C. sinensis* plant grafted on the four different rootstocks tested in semi-field experiments is reported in Figure 1, Appendix A, and Table 1. Overall, a marked different preference response was elicited by *A. gossypii* individuals that diversely infested the different targeted plants (Appendix A). The two-way ANOVA underlined that the two main effects (scion/rootstock combinations and times) were statistically significant, but their interaction was not significant (Table 1).

In each of the three timepoints, it was observed that aphids infested a mean (±SE) of 6.33 ± 1.88 leaves in plants of *C. sinensis* grafted on *C. aurantium*, a value significantly higher than that found in *C. sinensis* grafted on Carrizo citrange, where only 1.58 ± 0.57 leaves evidenced the presence of aphids (*p* < 0.01; ANOVA followed by Tukey’s test), and on Forner-Alcaide no. 5 and Volkamer lemon with, respectively, 2.33 ± 0.65 and 2.33 ± 0.61 leaves infested (*p* < 0.05; ANOVA followed by Tukey’s test). No statistical differences were recorded in the number of aphid infested leaves between *C. sinensis* grafted on Carrizo citrange, Forner-Alcaide no. 5, and Volkamer lemon (Figure 1; Appendix A).

### 2.2. VOC Chemical Analysis

VOC collection with headspace Solid-Phase Micro-Extraction (SPME) method followed by GC-MS analysis revealed the presence of a total of fifty-six VOCs released from the *C. sinensis* leaves grafted on the four different rootstocks. Ninety percent of compounds extracted were identified, following the present distribution: 17 monoterpene hydrocarbons, 5 monoterpene alcohols, 3 monoterpene aldehydes, 5 monoterpene esters, 1 monoterpene ketone, 15 sesquiterpene hydrocarbons, 2 aldehydes, and 2 green leaf volatiles (an alcohol and an ester) (Table 2).

The leaf VOC emission recorded underlined quantitative differences across the different *C. sinensis* combinations investigated. The two first components in the PCA analysis showed around 89% to the overall variability among the four scion/rootstock combinations analyzed (see Material and methods section), with a strong separation among them (Figure 2). Most of the compounds extracted were positively correlated with the first component (Dim1), while five compounds, three sesquiterpenes, one ester, and one alcohol (*Z*)-3-hexenol, the only one compound recorded in this category; Table 2), respectively, were negatively correlated. A clear discrimination across combinations according to their rootstocks and susceptibility to CTV infection was highlighted (Figure 2). Indeed, the four samples are located in four different quadrants: the *C. sinsensis* samples (CS) grafted on CTV-tolerant rootstocks obtained from trifoliata parents, Carrizo citrange (CC) and Forner-Alcaide no. 5 (FO), were in the right quadrant, in the upper and lower one, respectively, while the samples grafted on Volkamer lemon (CV), and on the CTV-susceptible rootstock *C. aurantium* (CA), were in the upper left and lower left quadrant, respectively. Fifty-five metabolites, about 90% of the total extracted, were shared between FO and CC (Figure 2; Table 2). Interestingly, two sesquiterpenes, *α*-farnesene and cis-*β*-farnesene, were correlated to the quadrant where CA was located, while *β*-sesquiphellandrene, hexenyl-acetate, and (*Z*)-3-hexenol were found in the CV’s quadrant (Figure 2).

A heatmap based on metabolite profiles isolated was able to separate the four combinations in two main distinct groups (Figure 3). The first cluster enclosed the samples grafted with two CTV-tolerant genotypes CC and FO, while in the other one, in two different branches, clustered the samples grafted on CA and CV, the CTV-susceptible and -tolerant samples, respectively (Figure 3). Limonene, sabinene, carene, trans-*β*-ocimene, linalool, and citronellal, all compounds belonging to monoterpene class, were the compounds showing the highest difference in the amount across samples (Figure 3).

PCA and heatmap analyses were able to underline a panel of compounds (sabinene, carene, limonene, trans-β-ocimene, linalool, citronellal, camphene, β-pinene, hexenyl-acetate, cis-β-farnesene, and α-farnesene) showing the highest discrimination power among combinations. ANOVA followed by Tukey’s test carried highlighted that CV + CS emitted a significantly lower amount of sabinene (F = 3.61; df = 3; *p* < 0.05) and trans *β*-ocimene (F = 4.29; df = 3; *p* < 0.05) in comparison to the other combinations. Finally, comparing the VOC profiles obtained from *C. sinensis* leaves of the four combinations investigated here (CA + CS, CC + CS, CV + CS, and FO + CS) to the compounds extracted from the leaves of each rootstock used (CA, CC, CV, and FO), previously reported in Guarino et al. [16], a modulation of rootstocks on the products emitted from *C. sinensis* (CS) was highlighted (Figure 4). Indeed, an overlap between the profiles of rootstocks and their combinations with CS was underlined (Table 2), but without a correlation in the amount of molecules emitted. Among combinations, CA + CS showed the highest level (40.3%) of shared compounds with its rootstock, followed by CV + CS (36.1%), FO + CS (23.4%), and CC + CS (22.2%) (Figure 4).

## 3. Discussion

The finding of tolerant alternative citrus rootstocks to Citrus Tristeza Virus (CTV), maintaining an adequate high yield and quality guaranteed by *Citrus aurantium*, is a priority to manage and overcome this disease and/or its vectors.

In the present study, it was found that CTV-susceptible and -tolerant rootstocks in *C. sinensis* scions elicited a different VOC profile from leaves and determined a specific attraction response toward *A. gossypii*, the main virus vector in Mediterranean areas, with an induction of avoidance mechanisms exhibited by CTV-tolerant rootstocks Carrizo citrange, Forner-Alcaide no. 5, and Volkamer lemon.

Recently, other studies have attempted to evaluate host preference in terms of the feeding and oviposition of other disease vectors toward un-grafted different citrus rootstocks. For example, it was observed that the psyllid *Trioza erytreae* (Hemiptera: Triozidae), a vector of Huanglongbing (HLB), prefers to orient toward Carrizo citrange, while it avoids Forner-Alcaide no. 5 [32]. Urbaneja-Bernat and co-workers [33] indicated that the Carrizo citrange rootstock is also highly vulnerable to the psyllid *Diaphorina citri* Kuwayama (Hemiptera: Liviidae) in comparison with other HLB vectors. Unlike the previous studies reported, our observations were carried out on grafted plants, the stage more likely to be attacked by the CTV vectors.

The behavioral observations on *A. gossypii* indicated that the aphids prefer to orientate and develop more frequently toward *C. sinensis* plants grafted onto *C. aurantium* rather than the other rootstocks. Due to the marked differences in the VOCs profiles of scion/rootstock combinations investigated, it is possible to argue that such variations could be the basis of the avoidance mechanism (antixenosis) observed toward the aphid. However, we cannot establish if other morphological or physiological factors might concur in the observed response. Indeed, our behavioral observations were carried out in semi-field conditions; therefore, the aphids were exposed to olfactory, visual, and morphological cues rather than only to the plant’s VOCs through an olfactometer instrument.

The relationships between aphids and their host plants are complex. In fact, the recognition of a plant as a suitable host and subsequent feeding initiation by an aphid depends on a composite interaction between aphid and plant traits [34]. This process of selection, settling, and feeding establishment on a new host plant is critical from the perspective of an aphid’s fitness, but is also a focal point in the evolution of plant resistance toward piercing-sucking pests [34,35]. Plants rely on unique mechanisms of recognition, signaling, and defense to cope with the specialized mode of phloem feeders as aphids [34]. The ability to locate and recognize host plants is essential for the survival of aphids, and this ability can be mediated by several factors, and among them, olfactory cues play a key role [36]. In particular, it has been observed that winged aphids exploit both visual and volatile cues to detect potential host plants [37], and the landing responses may be elicited by plant volatiles detected by antennal olfactory sensilla [38,39,40].

After settling on a host plant, aphids may assess its suitability based on surface molecules, including lipids and secondary metabolites [41]. Aphids then broadly probe and salivate into the plants, and often reject non-hosts after initial sampling of epidermal cell contents, or subsequent sampling of mesophyll cell contents [42]. Previous studies suggested that aphids do not show clear discrimination between host and non-host plants until they have landed and inserted the stylets [43,44]. Powell and colleagues [42] observed that, even if field and laboratory studies indicate that the major features influencing plant preference by aphids are perceived after stylet insertion, the cues detected by aphids before stylet insertion undoubtedly play a key role in the host selection process. Over the past decades, aphids have also been shown to acquire olfactory information on plant quality in contrast to in flight or when walking toward a plant, so host selection does indeed already begin pre-alighting [37,45]. In some cases, individual VOCs are used by aphids as host cues, and in others, specific VOCs act as non-host cues during host finding; for example, isothiocyanates can attract brassica-related aphids and discourage the other species [45,46,47]. However, in the majority of cases, the relative concentration of chemicals in a mixture of VOCs influences the host plant searching behavior of aphids, rather than one or two VOCs acting as key attractive compounds [48,49,50]. In fact, past studies in which insects have been exposed to plant-produced chemicals alone or in combination have showed that appropriate blends or combinations of VOCs determine stronger behavioral responses than single compounds [49,50].

In accordance, the data obtained in our study can suggest that the changes in the ratio of the VOCs emitted from different scion/rootstock combinations might influence the aphid preference behavior toward plants grafted on *C. aurantium*. In fact, the VOC profiles recorded from the leaves of *C. sinensis* grafted on the four different rootstocks showed differences in the compounds produced, mainly quantitative. Interestingly, among the compound observed, the volatiles belong mostly to terpenes, in agreement with the finding reported by Guarino and co-workers [16] from the rootstocks’ leaves investigated here. Similarly, a previous study [51] analyzing citrus leaves obtained from the Sugar Belle mandarin hybrid on three different rootstocks suggested that rootstock choice influences the overall scions VOCs profile, and this can enhance its defense by repelling vectors or for signaling to their natural enemies or parasitoids. In addition, in agreement with our findings, the differences were mainly related to the monoterpenes [51].

A possible explanation of aphids’ preference orientation toward plants grafted on *C. aurantium* could also be linked with their ability to use plant VOCs to discriminate between the suitability of different plants within the same species [45]. In fact, *C. sinensis* plants grafted on sour orange might have a higher vigor and a consequently higher number of young sprouts and leaves, the tissue mainly attacked by aphids, rather than the plants grafted on the other rootstocks tested in this study. The incidence of *A. gossypii* can be influenced by anatomical traits such as the age of shoots, or metabolic profiles, which can influence plant-insect-natural enemy interactions [50,51,52,53,54].

The higher presence of some VOCs in *C. sinensis* plants grafted on CTV-tolerant rootstock rather than the ones grafted on sour orange could suggest that such chemicals might concur in the avoidance mechanism observed toward *A. gossypii*. Sabinene, less abundant in the VOCs of the *C. sinensis* plant grafted on Volkamer lemon, is reported in the essential oils of *Schinus terebinthifolius* Raddi leaves, exhibiting anti-acetylcholinesterase activities toward the related *Aphis nerii* Boyer de Fonscolombe [55]. As mentioned above, we cannot establish if the avoidance mechanism exhibited by *A. gossypii* toward plants grafted on the CTV-tolerant rootstocks is determined by the not-landing of the individuals on the plants (antixenosis) or by their escape after probing/feeding of the plants (antibiosis). It is possible that both mechanisms can occur in synergistic combination as evidenced in *Brassica juncea-fruticulosa* toward the aphid *Lipaphis erysimi* (Kaltenbach) [56].

Finally, the possibility that the three CTV-tolerant species can emit secondary metabolites that deter aphids upon their feeding activity cannot be excluded and can be the object of further investigations. In the case of the same *A. gossypii,* it was observed that after their herbivory activity on *Gossypium hirsutum* L., the host plant produces a blend of defense VOCs including (*Z*)-3-hexenyl acetate, (*E*)-4,8-dimethyl-1,3,7-nonatriene (DMNT), methyl salicylate, and (*E*,*E*)-4,8,12-trimethyl-1,3,7,11-tridecatetraene (TMTT) that repel the cotton aphid, determining an induced antixenosis mechanism [57].

## 4. Materials and Methods

### 4.1. Plant and Insects

Two-years-old plants of *C. sinensis* (CS) grafted on *C. aurantium* (CA), Volkamer. lemon (CV), Carrizo citrange (CC) (*C. sinensis* × *P. trifoliata*), and Forner-Alcaide no. 5 (FO) were provided from Vivai Maimone Giuseppe Alessio located in Milazzo (Messina—Italy). Citrus plants were produced and maintained in a greenhouse at 25 ± 5 °C and 50–70% RH under natural photoperiod. Four plants of *C. sinensis* per different rootstock were grown on a substrate consisting of sand and peat (1:1) in 20 L cylindrical plastic containers and were watered 3 times per week and fertilized using alternating ratios of 3.1.1 and 1.3.1 (N.P.K). One month before starting the experiments, the plant material was transferred to IBBR Institute (Palermo-Italy) and maintained in shadow recovery condition until being used for collection and analysis of plant secondary metabolites.

An *A. gossypii* colony was established from infested *Hibiscus rosa-sinensis* L. plants. Insects were maintained in an environmentally controlled room (28 ± 2 °C, 70 ± 10% relative humidity, photoperiod 16L:8D) in wooden cages (25 × 25 × 40 cm) with two 5 cm diameter mesh-covered holes for ventilation. Aphids used in the experiments were both winged and wingless, mixed-instar individuals and were collected from the cultures immediately prior to bioassay.

### 4.2. VOC Chemical Analysis

Chemical analysis of the plant secondary metabolites for *C. sinensis* grafted on the different rootstocks (see “4.1 Plant and Insects” section) was carried out through the headspace SPME method [58], followed by gas chromatography and mass spectrometry (GC-MS). VOCs emitted from CS grafted on CA, CV, CC, and FO were separately collected in the headspace by using SPME. The stationary phase used as the coating was polydimethylsiloxane (PDMS, 100 μm) (Supelco, Bellefonte, PA, USA). A manual SPME holder from the same manufacturer was used for injections. Fibers were conditioned in a gas chromatograph injector port as recommended by the manufacturer at 250 °C for 30 min.

For the headspace collections, a one-year-old leaf chosen randomly from each plant grafted on the different varieties investigated (CA + CS, CC + CS, CV + CS, and FO + CS) was cut at the base of the petiole and immediately covered with parafilm to minimize the VOC emission from the point of cutting. Leaves were then gently placed by forceps into 22 mL glass vials, weighted using a precision balance, and sealed with a polytetrafluoroethylene silicon septum-lined cap (Supelco, Bellefonte, PA, USA). Subsequently, an SPME needle was then inserted through the septum and volatiles were absorbed on the exposed fiber for 5 min at controlled room temperature (22 ± 1 °C). Therefore, the experiments were carried out using four biological replicates by sampling leaves each from individual trees of *C. sinensis* grafted on the four different rootstocks (CA + CS, CC + CS, CV + CS, and FO + CS), for a total of 16 samples (4 plants x 4 rootstock).

In order to perform the chemical analysis of the collected VOCs, the loaded fiber was desorbed in the gas chromatograph inlet port for 1 min immediately after the end of the sampling time. Coupled GC-MS analyses of the headspace collections from the four plant species were performed on an Agilent 6890 GC system interfaced with an MS5973 quadruple mass spectrometer equipped with a DB5-MS column in splitless mode. Injector and detector temperatures were 260 °C and 280 °C, respectively. Helium was used as the carrier gas. The GC oven temperature started at 40 °C and then increased by 10 °C/min to 250 °C, with initial and final hold times of 5 and 20 min, respectively. Electron impact ionization spectra were obtained at 70 eV, recording mass spectra from 40 to 550 amu. Peak integration was carried out using ChemStation software.

### 4.3. Behavioral Bioassays

To investigate the *Aphis gossypii* preference response to *C. sinensis* grafted on the four different rootstocks, a semi-field bioassay was carried out in an experimental cage. The cage setup for the experiment, size 2.00 × 1.05 × 1.60 m, was made by a wood structure covered with a tissue-not tissue 17 g/mq (VERDEMAX^®^, Boretto (RE)—Italy). Sixteen plants, four per rootstock, were used by dividing them into four blocks with one treatment per block with a distance of 20 cm to each other (Appendix A), while, inside the block, the distance between the pots was 10 cm. Each plant was randomly placed in each block and their position followed a ten-days clockwise rotation.

At the start of the experiment and every ten days thereafter, cohorts of mixed-stage aphid individuals (approx. N = 100) were placed at the center of each block. Plants were then inspected every ten days (three samplings in total) to count the number of leaves per plant infested by *A. gossypii* individuals. After each inspection, aphids were not removed from the infested leaves, to resemble more realistically a field condition. The experiment lasted 30 days, from 15 May to 14 June 2022.

### 4.4. Statistical Analysis

VOC profiles recorded from each scion/rootstock combination were used to develop a PCA (Principal Component Analysis) using the R package FactoMiner [59] and factoextra (https://CRAN.R-project.org/package=factoextra; accessed on 13 July 2022). PCA allowed us to investigate what variables were able to separate the four samples investigated. A heatmap with the metabolites profiles for each variety was carried out by heatmap.2 in R package gplots (https://github.com/talgalili/gplots; accessed on 13 July 2022). The compounds that, in PCA and heatmap analyses, were able to discriminate the four combinations were further investigated by using a one-way ANOVA followed by Tukey’s test. A two-way ANOVA and Tukey’s multiple pairwise-comparisons were developed using R studio (https://www.rstudio.com; accessed on 13 July 2022) to verify the significance levels of behavioral bioassays results. Finally, a VENN diagram was also used to easily highlight the number of metabolites shared across *C. sinensis* grafted on 4 specific rootstocks studied here (see section “4.1 Plant and Insects”) and the VOC profiles previously extracted by Guarino and colleagues [16] from the rootstocks investigated. The diagram was developed using the R package VennDiagram (https://cran.r-project.org/package=VennDiagram; accessed on 13 July 2022).

## 5. Conclusions

This study showed that the *C. sinensis* plant grafted on CTV-tolerant rootstocks exhibited different VOC profiles in comparison with plants grafted on the CTV-susceptible sour orange. The behavioral bioassays with *A. gossypii*, the main CTV vector in Mediterranean countries, indicated that aphids prefer to orient and develop on plants grafted on CTV-susceptible rootstocks. This avoidance mechanism induced by the rootstocks Volkamer lemon, Forner-Alcaide no. 5, and Carrizo citrange could represent one of the co-factors at the basis of the tolerance toward the virus. Future studies will focus on the response of *C. sinensis* plants grafted on the CTV-tolerant rootstock after aphid feeding, through high-throughput approaches, also using other combinations with new promising alternative rootstocks [6], in order to provide useful solutions for the *A. gossypii* control and, consequently, the CTV disease in citriculture.

## Figures and Tables

**Figure 1 plants-11-03426-f001:**
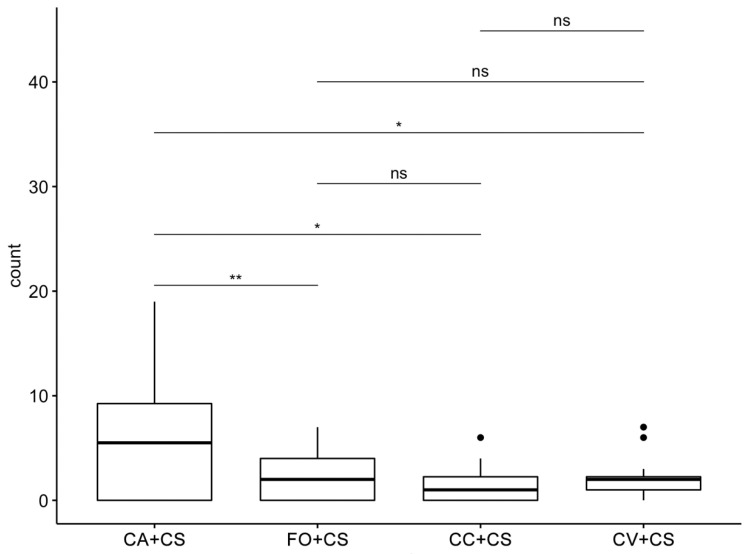
Preference response of *Aphis gossypii* toward the different combinations, i.e., number of leaves with presence of aphids per plant. Bioassays were carried out on the *Citrus sinensis* (CS) plants grafted on the four different rootstocks studied. CV: Volkamer lemon; FO: Forner-Alcaide no. 5; CC: Carrizo citrange; CA: *Citrus aurantium*. Asterisks indicate significant differences among the treatments (* *p* < 0.05; ** *p* < 0.01; ANOVA, followed by Tukey’s test). The values are reported in Appendix A.

**Figure 2 plants-11-03426-f002:**
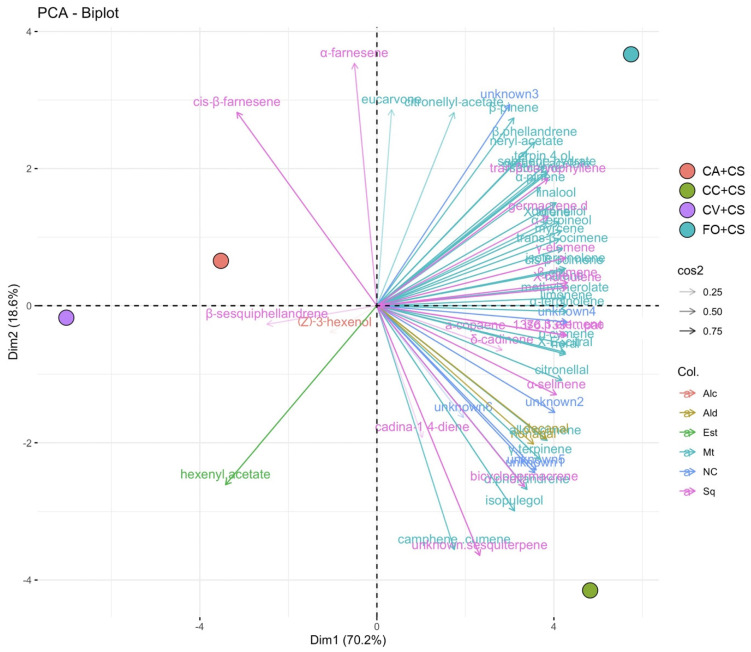
Biplot of the principal component analysis (PCA) for VOC profiles detected on leaves of *Citrus sinensis* grafted on four rootstocks showing different susceptibilities to CTV infection. Based on their profiles, samples were organized in four groups, and the associated compounds to varieties separation are indicated by vectors in the plot, underlining their significance values (0.25 < cos2 < 0.75). Each main category is highlighted with a different color: alcohol (Alc), aldehydes (Ald), esters (Est), monoterpene (Mt), and sesquiterpene (Sq). NC: not classified—unknown. CS: *Citrus sinensis*; CV: Volkamer lemon; FO: Forner-Alcaide no. 5; CC: Carrizo citrange; CA: *Citrus aurantium*.

**Figure 3 plants-11-03426-f003:**
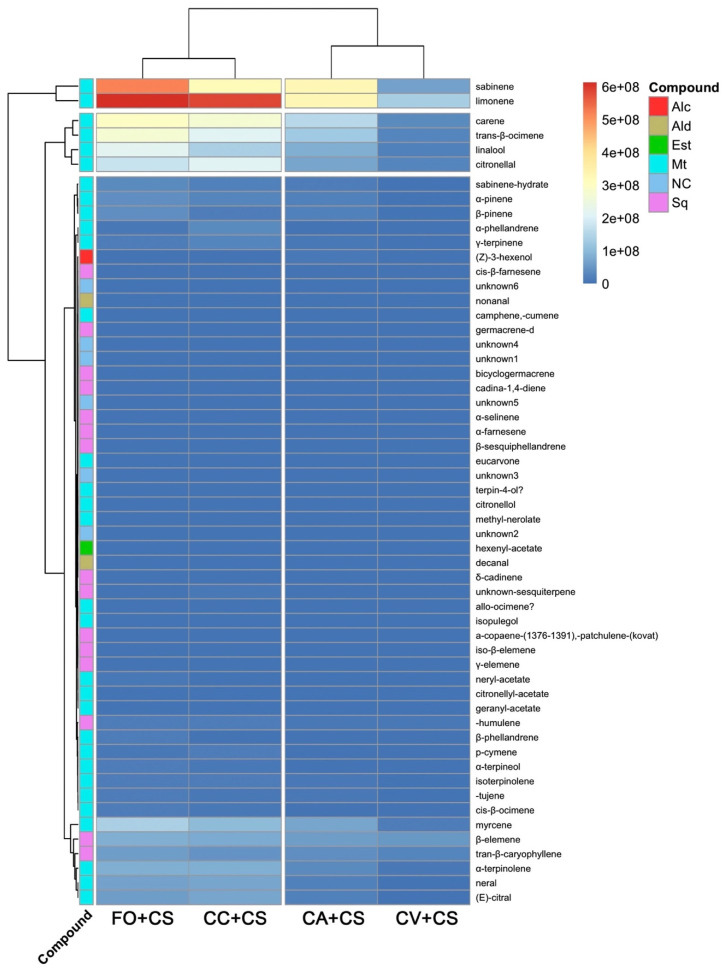
Heatmap on VOC emission of *Citrus sinensis* (CS) leaves grafted on Volkamer lemon (CV), Forner-Alcaide no. 5 (FO), Carrizo citrange (CC), and *Citrus aurantium* (CA). Alcohol (Alc), aldehydes (Ald) esters (Est), monoterpene (Mt), and sesquiterpene (Sq). NC: not classified—unknown. The heatmap legend (right) indicates the abundance (low—blue; high—red) of metabolites recorded in each sample.

**Figure 4 plants-11-03426-f004:**
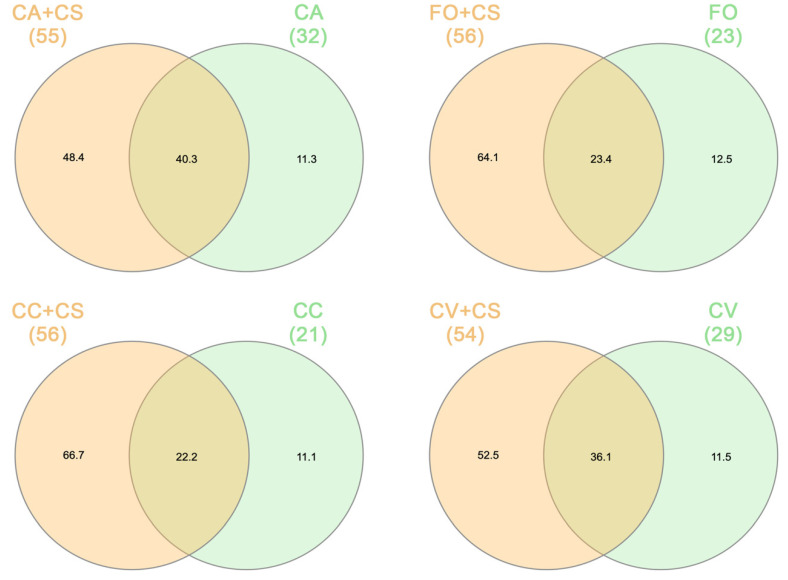
VENN diagram obtained by the VOC profiles comparison between the 4 rootstocks (CA, CC, CV, and FO) previously studied by Guarino et al. 2021 [16] and the compounds recorded in the *C. sinensis* grafted on the same 4 specific rootstocks investigated here (CA + CS, CC + CS, CV + CS, and FO + CS). The number of compounds extracted from each rootstock and combination was reported in bracket.

**Table 1 plants-11-03426-t001:** Behavioral response of *Aphis gossypii* toward the different combinations, i.e., number of leaves with presence of aphids per plant; two-way ANOVA with interaction effect across the scion/rootstock combinations.

	df	Sum Sq	Mean Sq	F Value	Pr(>F)
Scion/rootstock combinations	3	167.1	55.69	4.114	0.0131 *
Times	2	100.5	50.27	3.714	0.0342 *
Combinations:times	6	29.1	4.85	0.359	
Residuals	36	487.3	13.53		

* *p* < 0.05.

**Table 2 plants-11-03426-t002:** Volatile emissions ^1^ of *Citrus sinensis* (CS) leaves grafted on the different rootstocks *Citrus aurantium* (CA), Volkamer lemon (CV), Carrizo citrange (CC), and Forner-Alcaide no. 5 (FO) collected using headspace SPME. The compounds found also in the rootstock without scion reported in Guarino et al. [16] are in italic.

Peak	RT	LRI	Chemicals	Group	CA + CS	FO + CS	CV + CS	CC + CS
1	7.124	860	(*Z*)-3-hexenol	Alc. GLV	798.02 ± 460.62	19.46 ± 7.70	8.33 ± 5.89	220.77 ± 109.31
2	9.104	914	tujene	Mt. hd.	851.84 ± 380.29	1357.68 ± 385.62	111.53 ± 38.80	988.69 ± 337.15
3	9.277	925	*α-pinene **	Mt. hd.	2394.37 ± 760.63	3980.35 ± 1049.00	366.93 ± 123.02	2503.18 ± 742.30
4	9.72	942	camphene, cumene	Mt. hd.	71.86 ± 23.41	82.92 ± 37.30	91.99 ± 51.96	122.24 ± 54.47
5	10.259	968	*sabinene*	Mt. hd.	32,748.92 ± 8259.47	51,545.36 ± 11,420.14	6550.30 ± 2123.65	31,993.75 ± 8245.60
6	10.356	973	*β-pinene **	Mt. hd.	2293.41 ± 716.19	4070.04 ± 1037.37	357.42 ± 114.70	1669.62 ± 871.07
7	10.618	985	*myrcene **	Mt. hd.	6812.364 ± 2417.51	13,669.03 ± 3534.04	1269.10 ± 371.40	10,333.08 ± 2930.60
8	10.95	1001	hexenyl acetate	Est. GLV	183.64 ± 58.58	66.22 ± 33.11	206.27 ± 55.27	165.64 ± 69.95
9	11.032	1003	carene	Mt. hd.	15,815.20 ± 5681.08	30,026.01 ± 7916.07	3527.89 ± 916.41	27,053.19 ± 7171.47
10	11.208	1010	α-phellandrene	Mt. hd.	490.42 ± 350.66	1224.15 ± 387.80	59.52 ± 26.10	3092.84 ± 2440.88
11	11.371	1019	p cymene	Mt. hd.	391.90 ± 216.34	1141.53 ± 334.52	64.87 ± 30.64	1242.60 ± 410.88
12	11.499	1026	*limonene **	Mt. hd.	32,943.90 ± 13,842.73	61,289.52 ± 13,772.38	13,122.13 ± 4495.85	58,836.01 ± 14,022.15
13	11.569	1032	*cis β-ocimene*	Mt. hd.	600.04 ± 158.46	1307.78 ± 292.60	69.51 ± 35.51	1095.56 ± 226.63
14	11.789	1044	*trans β-ocimene*	Mt. hd.	12,706.48 ± 2560.20	27,137.86 ± 5385.58	2672.65 ± 735.35	21,118.00 ± 5553.26
15	11.935	1050	*β*-phellandrene	Mt. hd.	271.54 ± 183.16	1687.85 ± 999.24	42.87 ± 18.71	598.01 ± 228.26
16	12.024	1055	*γ-terpinene **	Mt. hd.	647.56 ± 304.32	1550.49 ± 511.75	72.35 ± 27.17	3026.11 ± 2282.45
17	12.255	1069	*sabinene hydrate*	Mt. est.	1289.81 ± 450.91	3572.42 ± 1270.21	195.13 ± 22.15	1872.28 ± 734.53
18	12.447	1078	isoterpinolene	Mt. hd.	702.78 ± 344.82	1659.58 ± 485.62	132.92 ± 37.86	1373.23 ± 451.32
19	12.524	1082	*α-terpinolene*	Mt. hd.	3630.55 ± 1735.17	8233.42 ± 2348.61	746.35 ± 214.08	7997.21 ± 2320.90
20	12.779	1097	*linalool **	Mt. alc.	8026.00 ± 2193.94	21,911.55 ± 6873.58	2222.82 ± 975.31	13,765.83 ± 4454.73
21	12.854	1099	nonanal *	Ald.	39.11 ± 23.29	87.98 ± 56.99	55.49 ± 22.70	121.32 ± 24.69
22	13.239	1127	*allo ocimene*	Mt. hd.	114.23 ± 21.13	310.39 ± 76.35	13.28 ± 9.39	537.47 ± 224.51
23	13.676	1151	*citronellal*	Mt. ald.	7293.26 ± 2811.10	17,251.92 ± 5625.10	2627.51 ± 226.28	21,392.37 ± 4304.05
24	13.825	1162	isopulegol	Mt. alc.	63.35 ± 30.39	116.99 ± 58.49	4.24 ± 2.99	383.29 ± 244.25
25	14.003	1171	unknown		28.44 ± 11.95	74.35 ± 37.17	9.26 ± 3.89	155.20 ± 79.58
26	14.096	1178	unknown		46.08 ± 33.98	189.40 ± 118.52	14.45 ± 5.65	278.03 ± 77.08
27	14.168	1181	*terpin 4-ol*	Mt. alc.	80.28 ± 24.17	337.22 ± 135.53	33.74 ± 13.55	163.17 ± 39.90
28	14.276	1186	unknown		0,00	233.99 ± 116.99	0	28.12 ± 20.14
29	14.393	1195	*α-terpineol **	Mt. alc.	276.33 ± 133.51	1247.17 ± 486.20	33.46 ± 16.47	823.98 ± 379.95
30	14.468	1199	unknown		32.82 ± 32.82	125.63 ± 50.95	13.37 ± 9.46	123.35 ± 51.03
31	14.509	1197	decanal *	Ald.	162.49 ± 26.79	235.78 ± 46.99	154.31 ± 33.48	299.19 ± 59.71
32	14.807	1224	*citronellol **	Mt. alc.	143.21 ± 90.79	293.54 ± 146.77	52.40 ± 33.07	212.47 ± 53.36
33	15.025	1239	*neral*	Mt. ald.	2156.31 ± 900.69	6235.03 ± 2596.47	338.38 ± 98.55	7071.17 ± 2316.41
34	15.205	1249	eucarvone	Mt. ket.	19.21 ± 15.21	269.63 ± 134.81	212.81 ± 97.08	41.04 ± 34.14
35	15.457	1268	*(E)-citral* *	Mt. ald.	2277.47 ± 1070.30	6024.86 ± 2599.98	103.01 ± 72.83	6853.64 ± 22.662
36	16.187	1317	methyl nerolate	Mt. est.	91.62 ± 43.20	194.71 ± 88.41	6.56 ± 2.33	181.06 ± 66.81
37	16.560	1344	citronellyl acetate	Mt. est.	74.58 ± 41.79	538.38 ± 269.19	301.74 ± 201.44	170.19 ± 58.38
38	16.688	1355	*neryl acetate*	Mt. est.	113.17 ± 54.55	830.55 ± 415.27	189.25 ± 121.52	354.29 ± 145.93
39	16.955	1375	*geranyl acetate*	Mt. est.	81.39 ± 69.73	496.57 ± 248.28	7.75 ± 3.84	228.30 ± 132.06
40	17.063	1382	*α*-copaene	Sq. hd.	348.76 ± 101.38	546.47 ± 141.59	257.51 ± 131.36	562.71 ± 147.59
41	17.122	1385	iso *β*-elemene	Sq. hd.	336.67 ± 131.31	457.75 ± 154.70	328.42 ± 221.38	460.94 ± 154.69
42	17.162	1388	unknown		10.78 ± 8.42	34.51 ± 17.25	0	75.33 ± 32.54
43	17.232	1393	*β-elemene*	Sq. hd.	5769.96 ± 1890.71	8054.76 ± 2419.72	4925.58 ± 3196.47	7561.09 ± 3158.02
44	17.343	1400	unknown sesquiterpene	Sq. hd.	61.42 ± 20.12	83.13 ± 30.69	57.29 ± 36.82	823.55 ± 753.85
45	17.693	1429	*tran β-caryophyllene **	Sq. hd.	3886.93 ± 1060.34	6016.15 ± 1780.72	2616.40 ± 1554.05	4416.97 ± 1462.25
46	17.787	1436	*γ-elemene*	Sq. hd.	330.60 ± 79.69	476.03 ± 181.18	271.92 ± 171.84	427.06 ± 163.10
47	17.964	1450	*cis β-farnesene*	Sq. hd.	376.12 ± 138.09	257.78 ± 110.69	342.80 ± 234.45	49.98 ± 28.85
48	18.026	1453	unknown		105.76 ± 30.95	120.34 ± 40.25	120.85 ± 71.58	125.56 ± 56.39
49	18.154	1464	humulene	Sq. hd.	1030.95 ± 295.69	1600.16 ± 463.95	750.07 ± 462.95	1489.90 ± 555.65
50	18.382	1481	*α*-selinene	Sq. hd.	23.60 ± 9.80	39.28 ± 19.64	7.20 ± 50.09	51.68 ± 20.39
51	18.464	1488	*germacrene d*	Sq. hd.	73.40 32.67	117.00 ± 32.61	77.29 ± 51.73	97.67 ± 63.23
52	18.617	1500	*α*-farnesene	Sq. hd.	66.42 ± 31.89	100.42 ± 48.89	92.41 ± 62.08	53.85 ± 47.60
53	18.644	1502	*bicyclogermacrene*	Sq. hd.	34.06 ± 19.24	45.69 ± 21.45	1.82 ± 1.28	110.71 ± 60.73
54	18.876	1521	*δ-cadinene*	Sq. hd.	160.88 ± 57.19	219.47 ± 86.81	200.84 ± 138.41	220.76 ± 108.37
55	18.926	1525	*β*-sesquiphellandrene	Sq. hd.	29.64 ± 21.64	45.365 ± 28.18	102.08 ± 72.181	45.12 ± 33.37
56	19.087	1538	cadina-1,4-diene	Sq. hd.	13.06 ± 5.55	26.80 ± 14.72	32.76 ± 23.16	34.44 ± 11.78

^1^ = Volatile emissions are given in mean peak area divided by 10^4^ with the ±SE. For all treatments, four biological replicates were carried out. * = Chemical compounds identified using synthetic standards.

## Data Availability

The data presented in this study are available in the text or Appendix A here.

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
