# Peer review of "Rootstocks with Different Tolerance Grade to Citrus Tristeza Virus Induce Dissimilar Volatile Profile in Citrus sinensis and Avoidance Response in the Vector Aphis gossypii Glover"

_plants, 2022, doi:10.3390/plants11243426_

Round 1
Reviewer 1 Report
The manuscript entitled “Rootstocks with different tolerance grade to Citrus Tristeza Virus induce dissimilar volatile profile in Citrus sinensis and avoidance response in the vector Aphis gossypii Glover” from Guarino et al. describes the leaf volatile organic compound (VOC) profiles from leaves of C. sinensis grafted on 4 different rootstocks (1 susceptible and 3 tolerant to CTV) as well as the behavior of the insect responsible for the disease transmission according to the scion/rootstock combination.
The MS is well written and well in the scope of Plants but needs some improvement:
1. Figure S1 legend is confusing (three times CA+CS) avoiding correct figure analysis.
2. Lines 98-104: please give the values for all the scion/rootstock combinations – not only for CS+CA (6.33±1.88) – allowing a better comparison between samples, and a better understanding of the Figure 1 (and also because such data are not shown anywhere – Table S1 showed statistics).
3. §2.2. The figure and table indications in the text are not correct (Figure 1, Table 1 instead of Figure 2 Table 2, etc.).
4. The VOC analysis (Table 2, Figure 2 and Figure 3) could be deeply explored regarding several points appropriately developed in the discussion but not in the result analysis such as:
a. The hypothesis was that CA is attractive for the insect – but what about the hypothesis that the other plant combinations may repelled it? What are the molecules that are more present in the FO/CV/CC plants?
b. The attractivity vs repelling could be due to a mixture of molecules. Try to better explore the heatmap in this sense (or use some statistical analysis from Table 2).
5. The Venn diagram comparing the present data with those of Guarino et al (2021) is interesting and smart, but the analysis would be deeper if the common vs exclusive molecules were listed and analyzed.
6. Figure 3 legend: ‘sequestered in the leaves’ corresponds to ‘VOC emission’?
7. What about the morphological and physiological stages/characteristics of the plants? The authors discussed the possibility that some other plant characteristics such as leaf thickness, color, etc. influence the attractivity of the insect. Please, give some comparative information about the plant morphology to support this discussion.
Author Response
The manuscript entitled “Rootstocks with different tolerance grade to Citrus Tristeza Virus induce dissimilar volatile profile in Citrus sinensis and avoidance response in the vector Aphis gossypii Glover” from Guarino et al. describes the leaf volatile organic compound (VOC) profiles from leaves of C. sinensis grafted on 4 different rootstocks (1 susceptible and 3 tolerant to CTV) as well as the behavior of the insect responsible for the disease transmission according to the scion/rootstock combination.
Thank you for the positive feedback and evaluation. We modified the manuscript following your suggestions. The changes were highlighted in red in the text.
The MS is well written and well in the scope of Plants but needs some improvement:
1) Figure S1 legend is confusing (three times CA+CS) avoiding correct figure analysis.
Response: sorry for the mistake. The legend in the figure S1 has been corrected
2) Lines 98-104: please give the values for all the scion/rootstock combinations – not only for CS+CA (6.33±1.88) – allowing a better comparison between samples, and a better understanding of the Figure 1 (and also because such data are not shown anywhere – Table S1 showed statistics).
Response: in accordance with the reviewer’s suggestion we pointed out all the values as requested, see lines 100-103
3) §2.2.The figure and table indications in the text are not correct (Figure 1, Table 1 instead of Figure 2 Table 2, etc.).
Response: sorry for the mistake. We corrected these oversights along the text in accordance with the reviewer’s suggestion.
4) The VOC analysis (Table 2, Figure 2 and Figure 3) could be deeply explored regarding several points appropriately developed in the discussion but not in the result analysis such as:
a. The hypothesis was that CA is attractive for the insect – but what about the hypothesis that the other plant combinations may repelled it? What are the molecules that are more present in the FO/CV/CC plants?
Response: Thank you for your suggestions. On the basis of PCA and heatmap analysis we selected a panel of compounds that discriminate the four combinations and they could be mainly associated to the repellent or attractive effect. The statistical analysis and the results were added in the manuscript in the appropriate sections.
b. The attractivity vs repelling could be due to a mixture of molecules. Try to better explore the heatmap in this sense (or use some statistical analysis from Table 2).
Response: thank you for your suggestion. Starting from available evidences already published (e.g. Bruce, et al. 2005, https://doi.org/10.1016/j.tplants.2005.04.003; Webster et al., 2010, https://doi.org/10.1016/j.anbehav.2009.11.028; Bruce and Pickett 2011, https://doi.org/10.1016/j.phytochem.2011.04.011) it is known that the attractive and repellent effects are often related to a mixture. We discussed this aspect in the discussion adding some more references and point out this concept more in depth, see lines 253-256. So, the present paper is a pivotal work on this interesting topic, the starting point. We are planning another manuscript with several activities to deeply investigate the effect of the molecules here extracted (including some mixture). The future activities will allow to validate the effect of compounds and to highlight the molecular pathways activates in plants (though high-throughput approaches) by the treatments.
5) The Venn diagram comparing the present data with those of Guarino et al (2021) is interesting and smart, but the analysis would be deeper if the common vs exclusive molecules were listed and analyzed.
Response: Thank you for your suggestion. We added the needed information in the Tables already included and in the text of the re-submitted version.
6) Figure 3 legend: ‘sequestered in the leaves’ corresponds to ‘VOC emission’?
Response, yes we changed in accordance with the suggestion of the reviewer to avoid confusion
7) What about the morphological and physiological stages/characteristics of the plants? The authors discussed the possibility that some other plant characteristics such as leaf thickness, color, etc. influence the attractivity of the insect. Please, give some comparative information about the plant morphology to support this discussion.
Response: Thank you for your suggestion. In the text we only hypothesize that the aphid fitness could be associated with the enhanced plant growth of specific citrus combination; this hypothesis is in accord to previous studies (e.g. Cornelissen et al. 2008, https://doi.org/10.1111/j.0030-1299.2008.16588.x; Che-Castaldo et al. 2019, https://doi.org/10.1002/ecm.1389; Saska et al. 2022, https://doi.org/10.1007/s10340-022-01514-3), that highlighted as the plant vigor can be directly influence the population dynamics, fitness, phenology, and biology of herbivorous insect. This assessment surely will be the focus of our next further study already planned. We added the correct references to support our hypothesis.
Reviewer 2 Report
GENERAL COMMENTS
The publication entitled: „ Rootstocks with different tolerance grade to Citrus Tristeza Virus induce dissimilar volatile profile in Citrus sinensis and avoidance response in the vector Aphis gossypii Glover” (Manuscript ID: plants-2069925) is interesting.
In my opinion, the manuscript is very valuable. It touches on very important aspects of the activity of Citrus Tristeza Virus (CTV) as a factor causing devastating epidemics of citrus plants grafted on Citrus aurantium. It is true that the answer to the stated goal of the research was obtained, the task of which was to explain if the distinctive scion/rootstock combinations can induce a preference or avoidance behavior in A. gossypii, and if CTV-tolerant rootstocks can modulate the VOC emission in C. sinensis scions, inducing different profiles in comparison to the susceptible sour orange. However, further detailed research is needed, as the authors emphasize in their conclusions, so that the research results can be confidently implemented.
The introduction is quite well written. The research site is also interesting. The authors used a correct research methods and received many results which were interpreted and statistically developed. An interesting chapter, which increases the substantive value of the manuscript, is the "Discussion" chapter.
SPECIFIC COMMENT
Chapter „Introduction”
I suggest adding information in the introduction to the question: Is it possible to recognize early enough symptoms of infection to be able to use other solutions apart from the washer? And what countermeasures have been used so far?

Author Response
The publication entitled: „ Rootstocks with different tolerance grade to Citrus Tristeza Virus induce dissimilar volatile profile in Citrus sinensis and avoidance response in the vector Aphis gossypii Glover” (Manuscript ID: plants-2069925) is interesting.
In my opinion, the manuscript is very valuable. It touches on very important aspects of the activity of Citrus Tristeza Virus (CTV) as a factor causing devastating epidemics of citrus plants grafted on Citrus aurantium. It is true that the answer to the stated goal of the research was obtained, the task of which was to explain if the distinctive scion/rootstock combinations can induce a preference or avoidance behavior in A. gossypii, and if CTV-tolerant rootstocks can modulate the VOC emission in C. sinensis scions, inducing different profiles in comparison to the susceptible sour orange. However, further detailed research is needed, as the authors emphasize in their conclusions, so that the research results can be confidently implemented.
The introduction is quite well written. The research site is also interesting. The authors used a correct research methods and received many results which were interpreted and statistically developed. An interesting chapter, which increases the substantive value of the manuscript, is the "Discussion" chapter.
Thank you for the positive feedback and evaluation.
SPECIFIC COMMENT
Chapter „Introduction”
I suggest adding information in the introduction to the question: Is it possible to recognize early enough symptoms of infection to be able to use other solutions apart from the washer? And what countermeasures have been used so far?
Response: Thank you for your comments. The earliest visual symptoms, such as vein clearing and stem pitting, are highly specific to the CTV patho-system; however, unfortunately, other symptoms (vein flecking, stunting, slow decline, and quick decline) are often mistaken with other diseases and nutrient deficiency. Therefore the only way to have an early diagnosis is a molecular approach (e.g. ELISA or qPCR).